# Taking Advantages of Blood–Brain or Spinal Cord Barrier Alterations or Restoring Them to Optimize Therapy in ALS?

**DOI:** 10.3390/jpm12071071

**Published:** 2022-06-29

**Authors:** Hugo Alarcan, Yara Al Ojaimi, Debora Lanznaster, Jean-Michel Escoffre, Philippe Corcia, Patrick Vourc’h, Christian R. Andres, Charlotte Veyrat-Durebex, Hélène Blasco

**Affiliations:** 1Laboratoire de Biochimie et Biologie Moleculaire, CHRU Bretonneau, 2 Boulevard Tonnellé, 37000 Tours, France; patrick.vourch@univ-tours.fr (P.V.); christian.andres@univ-tours.fr (C.R.A.); charlotte.veyratdurebex@univ-tours.fr (C.V.-D.); helene.blasco@univ-tours.fr (H.B.); 2UMR 1253 iBrain, Université de Tours, Inserm, 10 Boulevard Tonnellé, 37000 Tours, France; yara.alojaimi@univ-tours.fr (Y.A.O.); debora.lanznaster@univ-tours.fr (D.L.); jean-michel.escoffre@univ-tours.fr (J.-M.E.); philippe.corcia@univ-tours.fr (P.C.); 3Service de Neurologie, CHRU Bretonneau, 2 Boulevard Tonnellé, 37000 Tours, France

**Keywords:** blood–brain barrier, blood spinal cord barrier, amyotrophic lateral sclerosis, drug design

## Abstract

Amyotrophic lateral sclerosis (ALS) is a devastating neurodegenerative disorder that still lacks an efficient therapy. The barriers between the central nervous system (CNS) and the blood represent a major limiting factor to the development of drugs for CNS diseases, including ALS. Alterations of the blood–brain barrier (BBB) or blood–spinal cord barrier (BSCB) have been reported in this disease but still require further investigations. Interestingly, these alterations might be involved in the complex etiology and pathogenesis of ALS. Moreover, they can have potential consequences on the diffusion of candidate drugs across the brain. The development of techniques to bypass these barriers is continuously evolving and might open the door for personalized medical approaches. Therefore, identifying robust and non-invasive markers of BBB and BSCB alterations can help distinguish different subgroups of patients, such as those in whom barrier disruption can negatively affect the delivery of drugs to their CNS targets. The restoration of CNS barriers using innovative therapies could consequently present the advantage of both alleviating the disease progression and optimizing the safety and efficiency of ALS-specific therapies.

## 1. Introduction

Amyotrophic lateral sclerosis (ALS) is a neurodegenerative disease characterized by the degeneration of both upper and lower motoneurons. Patients usually die from respiratory failure after 3 to 5 years following the appearance of symptoms [1]. Several mechanisms contribute to the development and progression of ALS, including the aggregation and accumulation of ubiquitinated protein inclusions in motoneurons, alterations of mRNA processing, glutamate-mediated excitotoxicity, oxidative stress, mitochondrial dysfunction, and neuroinflammation [2]. Therapeutic options for ALS patients are very limited and mostly supportive and symptomatic. To date, only two drugs are FDA-approved for ALS: riluzole, which targets mainly the glutamatergic system, and edaravone, which targets oxidative stress. However, these molecules only modestly extend patient survival by a few months. Indeed, numerous drugs that targeted the main pathological mechanisms involved in ALS have been tested in clinical trials but failed to demonstrate a significant benefit in patients [3]. Failures of the numerous tested compounds can be explained in part by the choice of the target in regard to the complex pathophysiology of ALS, the small size of cohorts, and the heterogeneity of ALS patients. Despite continuous advances in drug discovery, the development of therapies targeting disease of the central nervous system (CNS) is complicated and limited by the presence of the blood–brain barrier (BBB) and/or blood–spinal cord barrier (BSCB). BBB/BSCB integrity has been rarely explored in ALS. Although some reports describe a disruption of these barriers, their role in the development and progression of the disease, or their potential consequences on drug delivery into the brain are still debatable. In this context, the objectives of this review are to shed light on BBB and BSCB alterations in ALS and their consequences on CNS-targeting therapeutics, to finally evaluate the interest of restoring the integrity of these barriers versus taking advantage of their alteration for drug administration. First, we will briefly describe the organization of normal BBB/BSCB and then focus on their alterations reported in ALS in regard to their association with the pathophysiology of the disease and their impact on drug pharmacokinetics. Therapeutic options to repair these barriers will be presented, as well as strategies that have been tested in ALS to overcome BBB/BSCB. Finally, we will discuss the interest of restoring its integrity to optimize the safety and distribution of a drug candidate designed to cross the BBB/BSCB versus considering BBB/BSCB disruption as an opportunity to reach the brain. 

## 2. The CNS Barriers: A Protection System

### 2.1. Organization and Functions of the Normal BBB and BSCB

There are three principal biological interfaces between the blood and the brain [4]. The first barrier is the BBB and the BSCB formed by endothelial cells (ECs). The second barrier is the blood–cerebrospinal fluid barrier (BCSFB) located at the epithelial cells of the choroid plexus. Finally, the arachnoid barrier surrounding the brain under the dura is avascular and presents a small surface area, which limits its role in the exchanges between the blood and the brain [5]. The BBB is the largest interface for these exchanges and therefore represents the most important barrier to prevent the entry of various substances into the brain, including drugs [4]. The composition of BBB, as well as BSCB, is complex and includes numerous entities forming the neurovascular unit (NVU): ECs, mural cells, basement membrane, astrocytes, immune cells and neurons [6] (Figure 1). ECs of the CNS have unique properties as they are joined by tight junctions (TJs), a complex of claudins and occludins linked regulatory proteins, which prevent the paracellular transport of numerous substances into the brain. Adherence junctions (Cadherin, JAMs) are essential to the structural support and the formation of TJs [4]. Mural cells are formed by pericytes and vascular smooth muscular cells that are distributed along the capillaries and partially surround the endothelium [4]. These contractile cells are essential to the BBB functionality as they can regulate the cerebral blood flow [6]. ECs and pericytes secrete and are enclosed by a basement membrane composed of a mixture of laminin, fibronectin and type IV collagen, which is essential to the maintenance of BBB integrity. Astrocytic endfeet form a complex network surrounding the capillaries, which help to the induction and maintenance of the barrier function. The complexity of these barriers ensures a variety of functions combining physical, transport and metabolic barriers [4]. These integrated elements protect the brain from the entry of neurotoxic compounds and are essential to the maintenance of a stable brain homeostasis. However, they also prevent the entry from the blood into the brain of most drugs and represent the major burden in the development of therapeutics for brain disease, including ALS [7].

The brain represents the main consumer of energy in the body. Additionally, while the BBB/BSCB appear to impassable obstacles, essential nutrients and metabolites can cross these barriers by several mechanisms, including passive diffusion, solute carrier transport, and vesicular transport. Immune cells from the circulation can enter into the brain by diapedesis (under inflammatory conditions) (Figure 1).

### 2.2. The CNS Barriers: A Burden for Brain-Targeted Therapeutics

Notably, very few drugs with a CNS target can enter the brain via the mechanisms previously described. Moreover, the detection of drug concentration into the CSF does not necessarily indicate a transport across the BBB but only across the BCSFB, much more permeable than the BBB [7]. It is important to keep this in mind as the diffusion of a molecule into the brain from the CSF will be limited near the CSF surface [7]. Multiple approaches have been developed to overcome the BBB for peripherally administered drugs targeting the CNS (Table 1).

#### 2.2.1. Mode of Administration

One strategy to bypass the BBB is the direct delivery of the drug into the CNS via intrathecal injection or convection-enhanced delivery. However, these methods are challenging and highly invasive. Moreover, they have limited diffusion near the injection site [9]. Bypassing the BBB can also be obtained by drug delivery to the brain through the nasal interface, where the molecule is deposited in the olfactory region and reaches the brain by crossing the olfactory epithelium [10].

#### 2.2.2. Engineering of Drugs

A second option is the re-engineering of the drug with fusion to a Trojan horse molecule, leading to a bifunctional entity. The molecular Trojan horse domain then binds to a BBB receptor such as the insulin receptor or Transferrin Receptor (TfR) to trigger BBB transport via receptor-mediated transcytosis. Drugs can also be combined to nanoparticles which can be organic (such as micelles, liposomes, or nanoemulsions) or inorganic (such as iron oxide or gold nanoparticles) [11,12]. These nanocarriers are able to be functionalized with agents targeting BBB components, with tracers to monitor drug distribution or with chemical compounds to activate the drug release via a stimulus [11].

#### 2.2.3. Permeabilization of the BBB

Other methods aim to transiently increase the permeability of the BBB either by using chemical strategies such as mannitol-mediated osmotic disruption and stimulation of bradykinin B2 receptor or by applying physical methods including radiation or microbubble-assisted focused ultrasound [10,13]. This last technique leads to local and reversible disruption of TJs. Modulation of active efflux transporter (P-gp, breast cancer resistance protein (BCRP), multidrug resistance proteins (MRPs)) by their direct inhibition or transcriptional repression is another possibility, but benefits are limited to the substrates of these transporters. 

There is no consensus about an optimal strategy to efficiently deliver substances into the CNS. An important point is that this optimal strategy must take into consideration the state of the BBB in the concerned CNS disease.

## 3. Strategies to Evaluate the BBB Integrity

Numerous direct and indirect methods have been reported to assess the functionality of the BBB/BSCB. Each method has its specific advantages and limits, and none is used consensually, which limits the comparisons between studies.

Postmortem histological observation of the BBB may provide molecular and ultrastructural information. However, the use of optical imaging (e.g., transmission electronic microscope (TEM), confocal or conventional optical microscopy) before/after tissue immunostaining requires the sacrifice of a large number of animals. In humans, this approach only reflects the end stage of ALS. The approaches based on peripherally administered tracers with various molecular weights such as sodium fluoresceine, fluorescent-labeled dextrans or Evans Blue can directly and quantitatively assess the BBB permeability, but these methods are only available in preclinical models and still require animal sacrifice. Moreover, no tracer presents optimal properties (non-toxic, not bound to other molecules, available in various molecular sizes, viewable and quantifiable) for an accurate determination of the BBB permeability [5]. 

Dynamic Contrast-enhanced Magnetic Resonance Imaging (DCE-MRI) can be used to quantify the regional BBB permeability (leakage of MR contrast agents) or to observe microhemorrhages in live patients. Positron emission tomography (PET), such as FDG-PET or Verapamil PET, is sometimes used but mainly inform about GLUT1 or P-glycoprotein (P-gp) functionality, respectively. Imaging techniques display the advantages to be performed in live individuals, allowing longitudinal monitoring, but require expensive and high-resolution equipment, especially if used on small animals. Moreover, these techniques are prone to inter-equipment variability (e.g., sensitivity) thus preventing the comparison between research groups.

Indirect markers of BBB/BSCB integrity can be quantified in biological fluids, mainly blood and CSF. These markers include the elevation of blood-derived molecules in the CSF such as total immunoglobulins, proteins, or albumin, but also more specific markers of neurological disease such as glial fibrillary acidic protein (GFAP), neuron-specific enolase (NSE), or S100 beta proteins. Ratio of molecules concentration between CSF and blood can also be used, the albumin quotient (QAlb) being the most routinely employed marker of BBB permeability. However, QAlb does not accurately reflect the BBB permeability, as albumin can be uptaken by the brain macrophages or glial cells and its CSF concentration depends on the fluid production or resorption [14]. It is generally advised to combine the determination of various blood/CSF ratios (e.g., IgG or α2macroglobulin quotients).

## 4. BBB Alterations in ALS and Their Consequences

Evidence of BBB alterations in ALS have recently been reviewed [15,16]. A summary of these alterations can be visualized in Figure 2. This section will focus on the role of these BBB alterations in the pathophysiology of ALS and their potential consequences on drug distribution in the brain, which are summarized in Table 2. 

### 4.1. Role of Alteration of the BBB in the Pathogenesis of ALS

#### 4.1.1. Findings from Animal Models

Structural and functional impairment of BBB and BSCB have been demonstrated in a *SOD1* mutated (G93A) mouse model of ALS through Evans blue leakage and the ultrastructure observation of the BBB components by TEM showed degenerated endothelial cells and astrocytes, extracellular edema or erythrocytes infiltration, which seem to worsen with disease evolution [17,18]. In various *SOD1* mouse models, Zhong et al. also observed signs of BBB alterations (reduction of TJ proteins level, hemosiderin deposits, reduction of capillaries length, and cerebral blood flow) in presymptomatic stages of the disease, before the detection of motoneuron loss and modification of inflammatory markers. These findings suggest that BBB damage might precede neurovascular inflammation and initiate symptoms [19]. A recent study also found signs of BBB alterations before the observation of neuromuscular denervation, within 30 days postnatal [20]. Moreover, TDP-43 overexpression obtained with intracranial injection of adeno-associated virus (AAV) vector containing the gene *TARDBP* in wild type mice was recently found to induce BBB permeability and inflammation leading to impaired motor learning, motoneuron loss, activation of astrocytes, and microgliosis [21]. The fact that the loss of BBB integrity promotes neuroinflammation, a key feature of ALS pathological mechanism [22], supports its implication in the pathophysiology of the disease. However, on the other side, in a *SOD1* mutated (G93A) rat model of ALS, Nicaise et al. observed signs of BBB alteration only at the symptomatic stage while IgG deposits were seen at presymptomatic stages, suggesting that BBB opening could be induced by pro-inflammatory cytokines [23]. The rare imaging studies conducted mostly on *SOD1* mutated rat models showed a positive correlation between infiltration of lymphocytes and gadolinium signal [24,25]. However, one study conducted on *SOD1* mutated (G93A) mice did not show elevation of Gadolinium leakage despite astrocytes activation and microgliosis, thus suggesting that BBB breakdown might not be a pathological aspect of all ALS cases [26]. Although decreased mRNA or protein levels of TJ proteins (mainly ZO-1, claudin 5, occludin) have been reported at diverse stages of the disease, TJs appeared structurally normal in TEM observation, both in animals and humans, which questions on the impact of their mRNA or protein downregulation on the paracellular pathway [17,23]

#### 4.1.2. Findings from Human Patients

Early evidence of BBB disruptions has been suggested with the observation of IgG and complement C3 deposits as well as lymphocyte and macrophage infiltration in the spinal cord or the cortex of postmortem ALS samples [27,28,29]. Some neuroimaging-based studies performed in ALS patients and regarding BBB permeability reported iron deposits, indicator of microhemorrhages and oxidative stress observed in ALS, but these findings remain debatable [30,31,32]. BBB opening using MR-guided focused ultrasound has been recently performed in four ALS subjects [33]. The gadolinium leakage normalized within 24 h, showing the reversibility of the procedure but also the absence of signs of previous BBB disruption in these patients. Moreover, elevation of QAlb has been reported in only 20–50% of ALS patients, suggesting that BBB disruption would not appear in all individuals [34,35,36]. Recently, Waters et al. showed that BSCB disruption, evidenced by hemoglobin leakage in postmortem human tissues, would be predominant in the thoracic spinal cord while motoneurons loss and TDP-43 deposits were mainly observed in the cervical and lumbar spinal cord. These data suggest that BSCB leakage in ALS is independent from motoneuron pathology [37]. Detection and evaluation of circulatory markers have almost exclusively been performed in human studies, as mouse CSF volume is very limited. Among the most recent ones, Li et al. compared several CSF parameters (total proteins, albumin, IgG, myelin basic protein) in addition to the QAlb and Quotient IgG (QIgG) between 113 ALS patients, 12 FTD-ALS patients, and 40 disease controls [38]. They found that CSF total proteins, CSF IgG, CSF albumin, QAlb and QIgG were significantly elevated in ALS patients. Moreover, CSF total protein, CSF IgG, and QIgG were significant indicators of disease progression. On the other side, Prell et al. evaluated the QAlb in a cohort of 160 ALS patients and 31 ALS mimicking conditions but did not find any significant association with the disease evolution [34].

#### 4.1.3. Further Necessary Investigations

In summary, little is known about the implication of BBB disruption in the pathogenesis of ALS. Histological evaluation in humans is limited to postmortem studies which prevents the estimation of the kinetics of BBB ultrastructure alterations. Therefore, animal models are essential to the longitudinal evaluation of BBB disruption. Despite a large variety of ALS animal models [39], studies evaluating BBB alterations focused almost exclusively on *SOD1* mutated rodents which validity is questionable. However, *SOD1* mutations represent only 1–2 % of sporadic ALS cases, and the number of genes associated with ALS has risen dramatically [40]. Even if QAlb is not a perfect marker of BBB integrity, the inconsistency of its elevation illustrates, once again, the complex heterogeneity of ALS, and suggests the implication of different pathophysiological mechanisms in different subsets of patients. It also highlights the limits of the different tools used to assess BBB integrity.

To date, studies failed to decipher a clear relationship between BBB disruption and pathological mechanisms. The beneficial effects of the restoration of BBB integrity in *SOD1* mutated mice, as discussed below, might support its implication in the pathogenesis of the disease, at least for this specific genetic background. A deep understanding of this phenomenon and its consequences will need: (1) structural and functional investigations in more animal models with various genetic background or even who mimics ALS pathogenesis [39] and (2) robust and accurate markers of BBB integrity routinely available in clinical practice.

### 4.2. Impact of the BBB Alterations on Drug Pharmacokinetics in ALS

Alterations of BBB ultrastructure and functionality would make this barrier leaky, as evidenced by the accumulation of blood-derived proteins or the infiltration of immune cells [15]. Moreover, permeability assays based on peripherally administered tracers show leakage of molecules with a wide range of molecular weights: from 376 Da sodium fluoresceine [41] to Evans Blue (representative of high molecular weight molecules permeability by binding to 65 kDa-albumin) [23]. This may suggest that a drug initially unable to cross the BBB (from small chemical molecules to high molecular weight antibodies) might still penetrate into the brain of ALS patients and reach its central target, without the need for a bypass strategy. However, it may also increase the cerebral toxicity via the entry of neurotoxic compounds, pathogens or therapies taken by the patient for other comorbidities. As clinical trials conducted on ALS patients continue to fail, this suggests that the tested molecules are either ineffective, or that they do not reach their target despite BBB alterations. 

#### 4.2.1. Upregulation of Efflux Transporters

Efflux transporters may explain the impossibility for a drug to reach its target. Indeed, BCRP and P-gp may be upregulated in ALS [15,42,43]. These transporters are widely expressed in various tissues and limit the absorption or accelerate the elimination of numerous conventional drugs, which may suggest their contribution to many treatment failures [44]. As an example, Riluzole is a substrate of these proteins, which may explain its modest efficiency in ALS. Jablonsky et al. observed an increased mRNA and protein expression of these transporters, as well as an elevated transport activity in symptomatic *SOD1 (G93A)* mutated mice [43]. They also reported an elevation of protein expression in a *TDP-43 A315T* mouse model and in ALS patients (but comparing only 3 patients to 2 controls) [43]. However, although these results have been successfully replicated for P-gp in other *SOD1* ALS models, this was not the case for BRCP [15]. An immunohistochemical evaluation of P-gp and BCRP in 25 ALS patients and 14 controls revealed a strong increase in these proteins in glial cells but not in blood vessels [45]. In 2019, Mohammed et al. found elevated protein expression of P-gp in human-iPS-derived ECs after co-culture with ALS human iPS-derived astrocytes [46]. Notably, P-gp was upregulated for *SOD1* and sporadic ALS derived astrocytes but not for *C9ORF72*, suggesting different alterations according to genetic background. Moreover, they showed that this upregulation of P-gp in ECs seemed mediated by glutamate release from astrocytes. 

#### 4.2.2. Drugs Diffusion into the Brain after Crossing the Barrier

When a drug succeeds in crossing the barrier formed by the ECs, it still needs to diffuse across the interstitial fluid to reach its target such as neurons. The distance between ECs and the neurons or glial cells is short, but some of the modifications observed in ALS might limit the diffusion of the molecule to its target. For example, accumulation of collagen IV has been reported in the brain or spinal cord of ALS patients [37,47]. In *SOD1 G93A* mice, collagen IV staining was progressively reduced in vascular structures but remarkably increased in tissues, as compared to non-transgenic mice [48]. Ultrastructure observations of the BBB in TEM also showed a thickening of the basement membrane in the brainstem, and in the cervical and lumbar spinal cord of *SOD1 G93A* mice [17]. The accumulation of collagen IV and the subsequent basement membrane thickening or blood-derived molecules deposition might form another barrier limiting the access of the candidate drug to its target cells [49]. This mechanism has been suggested to explain the reduced brain uptake of [^3^H] diazepam and [^3^H] propranolol in a mouse model of Alzheimer disease [50].

Moreover, TEM analyses also showed extracellular edema in spinal cord and brainstem in both animals and humans [17,47]. This phenomenon might also affect the penetration and distribution of the drug in the brain. In fact, Binder et al. reported that cytotoxic brain edema produced by water intoxication slowed the diffusion of fluorescein-dextran in the mouse brain and created dead-space microdomains in which free diffusion was prevented [51]. In this study, the deletion of aquaporin 4 (AQP4), a glial water channel enhanced the fluorescein-dextran diffusion in the extracellular space. As AQP4 was found upregulated in ALS [52,53,54], it could also be a limiting factor to the brain diffusion of candidate therapies.

#### 4.2.3. Limitation of Barrier Bypass Strategies by BBB Alterations

Most strategies to bypass the BBB rely on a healthy BBB, but its alteration in ALS could limit the efficiency of these strategies.

TfR is one of the most popular BBB receptors targeted in the molecular Trojan horse technology. However, iron metabolism is altered in ALS, and TfR appears to be dysregulated in different ways depending on the genetic background. Overexpression of wild type *SOD1* and *SOD1 G93A* but not *SOD1 H46R* mutation induced an increase in the protein expression of TfR in an in vitro experiment [55]. Another study found an elevation in the mRNA expression of TfR in *G93A-SOD1* cells, which is consistent with higher iron uptake [56]. On the other hand, mutations in *OPT* were associated with the degradation of TfR via autophagosomes, although this effect seemed limited to RGC-5 cells [57]. These reports suggest that the efficacy of a therapy using TfR targeted Trojan horse molecular strategy could vary in ALS. 

Prevention of drug penetration across the CNS by the accumulation of collagen IV and the thickening of the basement membrane may also limit the efficacy of bypass strategies such as intrathecal administration, nanoparticles, or BBB permeabilization [49]. Focused ultrasounds used for the delivery of therapeutic agents to the brain might need to be adjusted for application in CNS disorders. This hypothesis is illustrated in a preclinical mouse model of Alzheimer’s disease that displays a thickening of the basement membrane and where vessels were less permeable following focused ultrasounds application as compared to non-transgenic mice [58].

#### 4.2.4. Spatial and Temporal Alterations: A Source of Variability

As mentioned above, the kinetics of BBB disruption in ALS are unclear, as some studies reported signs of alteration at presymptomatic stages while others only after the appearance of symptoms. However, almost all studies which evaluated these alterations longitudinally agree that they tend to worsen as the disease progress. Thus, in addition to the inter-individual variability that seems to be reported for BBB disruption in ALS, there might also be an intra-individual variability that must be taken into account when considering the consequence of these alterations on the passage and diffusion of a candidate drug into the CNS. 

Moreover, the breakdown of the BBB is not uniform across the spinal cord and the brain. As previously discussed, Waters et al. evaluated the BSCB leakage of hemoglobin across the spinal cord of ALS patients and found that it was more severe at the thoracic level than the cervical or lumbar levels [37]. Moreover, collagen IV for example was only elevated in the white matter of the spinal cord but not in the gray matter. As TDP-43 inclusions and altered motoneurons are predominant in the cervical and lumbar spinal cord, this pattern does not correlate with the BSCB alterations. Therefore, the possible leakage into the brain of a drug candidate via BBB/BSCB disruption might not even benefit the cells that need it the most. The localization of primally affected motoneurons is different between patients with spinal onset (muscle weakness of the limbs) and patients with bulbar onset (dysarthria, dysphagia, speech difficulty) [2]. The localization of the increased BBB permeability reported in ALS might not be associated with that of affected motoneurons in spinal or bulbar forms. 

These phenomena illustrate the unpredictable pharmacokinetics of CNS targeted therapies in ALS, both among different patients and throughout the progression of the disease in the same patient, which might be hazardous for drugs with a narrow therapeutic index.

The consequences of BBB alterations on brain distribution of drug candidates are poorly understood. They might vary between subjects, throughout the evolution of the disease, across the different regions of the spinal cord or the brain. Such alterations might even limit the efficacy of a bypass strategy. Thus, monitoring and restoring the BBB integrity may be a valuable therapeutic strategy to optimize the administration of therapeutic molecules (Figure 3).

## 5. Therapeutic Strategies by Correcting BBB/BSCB Alterations

Approaches to restore or protect the BBB or BSCB in neurodegenerative or other diseases have already been reviewed elsewhere [14,59]. Briefly, therapeutic methods have focused on the BBB restoration with preservation of ECs and TJs, reducing the formation of edema by targeting AQP4, preventing the degradation of basement membrane by matrix metalloproteinases, elimination of neurotoxic deposits, enhancement of the clearance function and stem cell transplantation therapies to regenerate damaged tissues [14,59]. Here, we will focus on approaches that have been evaluated in ALS models and on recent advances in this field.

### 5.1. Previous Attempts in ALS 

Activated Protein C (APC) and analogs of this molecule have been evaluated in *SOD1* mutated (G93A) mice [60,61]. They reduced damage to the BSCB with a blockade of IgG and hemoglobin leakage and reduction of microhemorrhages (evidenced by hemosiderin staining) and free iron. The repair of the barrier function was probably due to the observed restoration of TJ proteins levels. Moreover, APC and its analog increased the lifespan of mice and the duration of the symptomatic phase. Notably, in one of these studies, another therapy evaluated (iron chelation) only alleviated a specific aspect of BBB disruption (iron deposit) with modest beneficial effects. Importantly, therapies given pre-symptomatically [60] extend more the lifespan as compared to post symptomatic treatment [61], meaning that early maintenance or repair of the BBB is more beneficial. These findings also support the role of the BBB disruption in the pathophysiology of the disease. Antagonizing the CXC4 receptor was also found to restore BSCB permeability and improve the survival of *SOD1 G93A* mice [62].

Stem cells transplantation for BBB restoration has been evaluated in vitro or in ALS preclinical models. Mesenchymal stromal cells and pericytes have been evaluated on presymptomatic *SOD1* mutated (G93A) mice [63]. Only pericytes had a slight significant effect on survival. However, no signs of BBB restoration were evaluated here. Other studies aiming to restore the BBB in ALS have been led by Garbuzova-Davis et al. Their goal was to restore the BBB alterations with peripheral administration of cells able to engraft themselves within the capillaries of the spinal cord and to differentiate into functional ECs. In various studies, they evaluated the effects of human bone marrow-derived CD34+ cells (hBM34+) or restricted-lineage endothelial progenitor cells: human bone marrow-derived endothelial progenitor cells (hBMEPCs) [64,65,66,67,68,69]. In summary, they found that both lineages ameliorated ALS outcomes and allowed for BBB repair. These findings were based on the following significant observations: reduction of Evans blue leakage, decreased astrocytosis and microgliosis, amelioration of capillaries ultrastructure, reduction of microhemorrhages, enhancement of basement membrane integrity, restoration of pericyte coverage, endothelial markers and TJ proteins. hBMEPCs displayed better engraftment and differentiation than hBM34+ cells, leading to improved outcomes and BBB integrity restoration. Recently, extracellular vesicles (EV) derived from hBMEPCs were beneficial to mouse brain ECs [70]. This illustrates the potential of EV-derived hBMEPCs as an innovative BBB-restoring therapy in ALS, but preclinical investigation is now necessary. 

### 5.2. Recent Advances with Direct and Indirect BHE Targeting

#### 5.2.1. Stem-Cell Therapies in Human

Stem-cell based therapies have been tested in ALS patients and demonstrated some benefits without substantial improvement in disease progression [71,72,73]. To our knowledge, no evaluation of BBB permeability following stem-cell based therapies was performed on patients. In adrenoleukodystrophy, a X-linked peroxisomal disorder, regions of demyelination are associated with gadolinium enhancement on MRI, sign of BBB disruption [74]. Hematopoietic cell transplant, the only therapy which stops the neurologic progression of the disease, allowed gadolinium resolution (indicator of BBB repair) within 100 days for almost all patients [75]. The underlying proposed mechanism is that donor mononuclear cells or microglia precursors would cross the BBB, differentiate into microglial cells, and attenuate neuroinflammation.

#### 5.2.2. Targeting Oxidative Stress and Inflammation

Caspase-1, a core component of the inflammasome complex is a promising therapeutic target in CNS disorders with neuroinflammation. Inhibition of caspase-1 reduced the transmigration of mononuclear cells across an in vitro BBB model exposed to an inflammasome-dependent pro-inflammatory response and restored the BBB integrity probably via the restoration of cadherin adherens protein [76]. Moreover, inhibition of caspase-1 also demonstrated its efficiency in vivo in various CNS disorders such as Alzheimer’s disease [77], Parkinson’s disease [78] and multiple sclerosis [79] which illustrates its potential as a target for BBB repair [80]. Utilization of inhibitors of caspase-1 may be a promising therapeutic approach via the restoration of BBB integrity, which could allow for a better control of therapeutic administration and an improvement of neuroinflammation-induced symptoms. Interestingly, melatonin, a non-specific inhibitor of caspase-1 activation has previously been administered in *SOD1* mutated mice and revealed an improvement in disease progression and an amelioration of motoneuron loss and spinal cord atrophy. However, its effect on BBB permeability were not evaluated [81].

In a mouse model of BBB disruption induced by traumatic brain injury, pharmacological elevation of nicotinamide adenine dinucleotide (NAD) restored the BBB integrity, as suggested by the reduction of IgG infiltration and the number of active microglia [82]. They also observed a restoration of endothelium length and an increase in capillary pericytes and TJ proteins. Moreover, this treatment restored the native BBB permeability to 3kDa dextran, initially enhanced after administration of Liposaccharides (LPS) and protected cultured human microvascular endothelial cells from oxidative stress. Target Nrf2 antioxydative signaling might also be a therapeutic target, as its stimulation by fenretinide protected the BBB against LPS in a mouse model [83]. Neutralization of the pro-inflammatory factor High mobility group box-1, by monoclonal antibody or inhibition of its release also showed a protective effect on BBB integrity [84].

#### 5.2.3. BBB Restoration in Combination of BBB-Opening Strategy

As mentioned above, acoustically mediated BBB opening using microbubble-assisted focused ultrasounds can locally and transiently increase the permeability of BBB for drug delivery. Normally, TJ proteins return to baseline levels within few hours following focused ultrasounds. However, this restoration might be impaired in neurodegenerative diseases with preexisting alterations in TJs, thus leading to a longer exposure of the CNS to neurotoxic blood components. For this matter, Lynch et al. explored the capacity of vasculotide, an angiopoietin 1 mimetic peptide, to restore BBB permeability after focused ultrasounds application [85]. Vasculotide was chronically administrated to a mouse transgenic model of Alzheimer’s disease. BBB restoration was assessed by longitudinal gadolinium enhancement after MRI-guided focused ultrasounds BBB-opening. Vasculotide significantly reduced BBB closure time (without alteration of the opening) and histological analysis (24 h post focused ultrasounds) revealed that the brains of treated animals displayed less infiltration of blood-derived molecules and cells (IgG, fibrinogen, erythrocytes). These results highlight the potential of vasculotide to accelerate BBB restoration after focused ultrasounds exposure with a potential subsequent improvement in safety and efficacy of treatments.

There is a continuous discovery of new approaches for the repair of BBB or BSCB integrity, but rare therapeutics have been successfully translated to humans. The more valuable option would probably be a combination approach to protect all components of the NVU.

## 6. Therapeutic Strategies to Overcome BBB in ALS

Numerous drugs that focus on main pathological mechanisms involved in ALS have been tested in clinical trials but failed to demonstrate a significant benefit in patients. However, it is difficult to conclude that a given biopharmaceutical is ineffective for a brain disease if the drug never reaches the target site. 

Using effective strategies to overcome this barrier and allow a drug to reach its target may give the opportunity to re-evaluate previous disappointing drugs. The benefits of riluzole in ALS might be limited by the upregulation of P-gp observed throughout the disease. In this context, alternative delivery systems have been evaluated in vitro to improve the benefits of this drug, mainly using nanoparticles formulations. Such alternatives include a liposomal formulation capable of co-delivering riluzole with a P-gp inhibitor, verapamil [86], and a lactoferrin-functionalized nanocarriers that allows the receptor-mediated transcytosis of riluzole across the ECs [87]. However, these promising results need to be confirmed in vivo. Other studies aiming to enhance brain distribution of riluzole in preclinical context include intranasal administration of riluzole-loaded nanoemulsion in rats [88], or intraperitoneal injection of riluzole-encapsulated nanoparticles [89,90]. Intracerebroventricular injection of another neuroprotective agent minocycline encapsulated in LPS-modified liposomes showed better efficiency than its conventional formulation in a *SOD1* mutated mouse model of ALS [91].

In recent decades, biopharmaceuticals revolutionized the treatment of a broad range of diseases and are increasingly used in many fields of medicine. However, the successful development of biopharmaceuticals is complicated by their limited brain access. Neurotrophic factors, for example, were rapidly tested in clinical trials in ALS but failed, probably because they were never delivered to the CNS [92]. Since then, bypass strategies to deliver therapies to their targets have been evaluated, including intranasal delivery in rats [93], the targeting of the Glial cell line-derived neurotrophic factor (GDNF) expression in the skeletal muscle via AAV vector coding for GDNF [94], or the transplantation of progenitor cells secreting GDNF [95,96,97]. This last strategy is retained in a Phase I clinical trial where the safety of neural progenitor cells secreting GDNF (CNS10-NPC-GDNF) transplantation in the spinal cord of ALS patients will be evaluated (ClinicalTrials.gov Identifier: NCT05306457). The discovery of new potential therapeutic targets leads to an increased number of emerging small molecule approaches which will require a good CNS penetration to be successfully transposed in humans [98]. As drug carriers or therapeutics drugs by themselves, nanoparticles could be a valuable option as evoked for the encapsulation of riluzole in organic nanoparticles. Cerium oxide inorganic nanoparticle administrated in SOD1 mutated mice showed benefit on the symptoms and the lifespan of the transgenic, probably by its antioxidant effects. This compound also displayed a penetration into the CNS as evidenced by cerium concentration into the brain [99].

Bypassing the BBB with direct delivery into the CSF via intrathecal or intracerebroventricular injection only allows the diffusion of the drug into the CNS near the injection site [9]. Indeed, diffusion decreases logarithmically with distance, which is problematic when the target is distant from the injection site. This could contribute to the difficulty of translating promising results found in animals to humans. For instance, intrathecal administration of an antisense oligonucleotide (ASO) targeting SOD1 (Tofersen) recently failed a phase III clinical trial [100], despite promising results in rodents [101] and signs of target engagement with a 30% reduction level of CSF SOD1 for the maximum tested dose (100 mg) in phase II trial [102]. A recent pharmacokinetic model was developed to describe ASO distribution after intrathecal injection based on previous non-human primate data [103]. It predicted that only 4 % of the injection dose would reach the CNS (which remains higher than following peripheral administration), in addition to local differences in the ASO concentrations along the spinal CSF canal, which may limit access to target tissues in humans. This illustrates that ASO therapy in ALS might need a better bypassing strategy than intrathecal administration. In this way, encapsulation of an ASO targeting *SOD1* in calcium phosphate lipid nanoparticles displayed uptake in an in vitro model of motoneuron-like cells (NSC-34) and were able to reach the brain and spinal cord after direct injection in a zebrafish model [104]. However, these results still need to be confirmed in mammalian models. Intrathecal administration of AAV allowing gene silencing in the brain or the spinal cord has been conducted in ALS (mostly targeting *SOD1*) with promising results in animal models and its potential was also demonstrated in two patients [105,106]. AAVs have also been used to deliver therapeutic transgene in specific cells such as insulin-like growth factor 1, GDNF (as mentioned above), hepatocyte growth factor or neuromuscular junction proteins [105]. As mentioned above, safety and feasibility of transitory opening of the BBB using MR-guided focused ultrasounds in ALS has been demonstrated by Abraho et al. [33]. After optimization in future trials, this method could represent a non-invasive opportunity to deliver various therapeutic agents into the CNS, including small molecules and nanoparticles. Moreover, targeting specific regions of the CNS with MR-guided focused ultrasounds could allow specific delivery of the therapeutic agents to the affected motoneurons according to the onset of the disease (spinal or bulbar). Notably, application of this method to the spinal cord will be challenging in primate vertebrae [107]. 

While strategies to overcome the BBB start to emerge in ALS, bringing new hopes for the disease, translating these bypassing strategies from in vitro or preclinical models to humans still need to prove its efficiency and will probably be hampered by the complex BBB alterations observed in ALS. 

## 7. Conclusions: Taking Advantage of BBB/BSCB Alterations or Restoring the Barriers to Optimize Therapy in ALS?

Despite continuous advances in the understanding of the pathological mechanisms of ALS, conventional pharmacological clinical trials failed to provide any efficient cure. The development of therapies for brain diseases is complicated and limited by the presence of CNS barriers. There is evidence of BBB and BSCB alterations in ALS but their roles in the complex pathogenesis of the disease remain poorly understood and require further investigation. We could suggest that these alterations provide an opportunity for a drug to penetrate into the brain, but the consequences on drug pharmacokinetics and pharmacodynamics are not obvious. First, BBB alterations do not seem to appear in all patients. Second, the upregulation of efflux transporters, formation of extracellular edema and the thickening of the basement membrane might limit the diffusion of the therapy across the interstitial fluid. Moreover, the unclear localization and progression of these alterations is a source of unpredictability and variability concerning the level of the drug that can actually reach its target. 

Given these arguments and continuous failures of therapies targeting the CNS over the past decades, we do not recommend assuming that BBB disruption in ALS will allow an optimal access of the candidate molecule to its target. Thus, it appears that efficient therapies for ALS will need the innovative development of an optimal strategy to overcome the transport across the BBB, as direct intrathecal or intracerebroventricular administration alone might be insufficient. However, as mentioned above, some alterations of the BBB could also limit the efficiency of such bypassing strategies. Restoring the BBB/BSCB could be very advantageous in ALS, because it could present the combined advantages of reversing a potential pathological mechanism of the disease, which still needs further investigation, and enhancing the safety and optimization of specific co-medication in ALS.

So, we suggest the following: (1) to find robust and non-invasive biomarkers of BBB/BSCB disruption to stratify patients according to the state of this barrier, (2) to repair these alterations for patients in which the bypass strategy might be less effective, and (3) to use innovative modalities to bypass the BBB, leading to a more personalized medicine approach based on the BBB/BSCB status. 

## Figures and Tables

**Figure 1 jpm-12-01071-f001:**
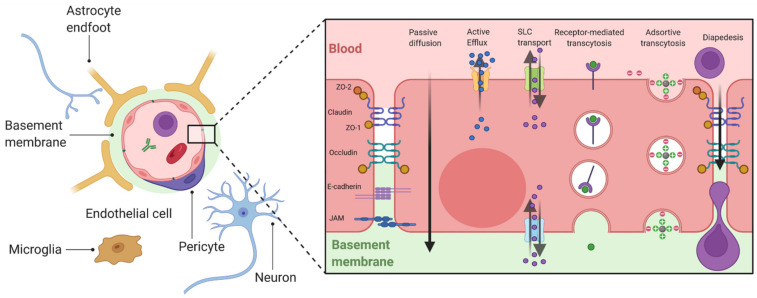
General organization of the BBB and mechanisms of transport. ECs are polarized cells which present at the apical or basolateral membrane numerous membrane transporters allowing bidirectional exchanges between the brain and the blood. Passive diffusion concerns dissolved gazes and small weight liposoluble molecules (generally <400 Da) [4]. The passage of these molecules can be limited by the fact than they can be substrates of apical efflux transporters, mainly belonging to the ATP Binding Cassette (ABC) family of transporters [8]. Polar nutrients may diffuse across the BBB but mainly enter the brain via carrier transporters such as the solute carrier transporters (SLC) family [5]. Larger molecules such as peptides or proteins can enter into the brain by vesicular transport, including receptor-mediated transport which involves endocytosis by the fixation of a ligand to a receptor (e.g., transferrin and its receptor (TfR)), and adsorptive-mediated transcytosis which concerns cationic molecules [4]. Created with BioRender.com.

**Figure 2 jpm-12-01071-f002:**
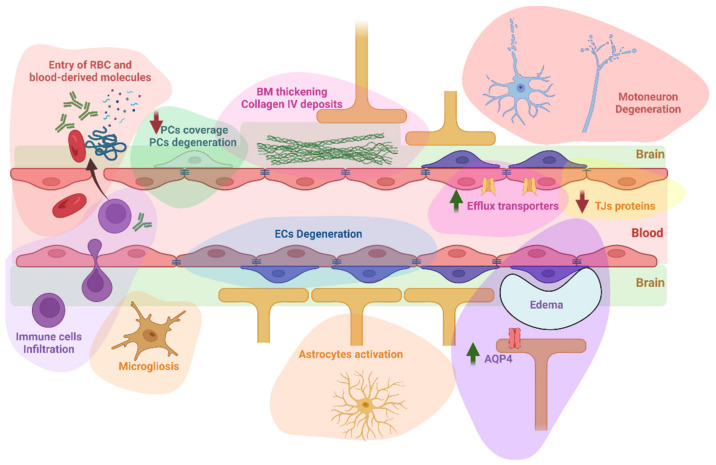
A summary of BBB/BSCB alterations reported in ALS. These alterations include the infiltration into the brain of circulating erythrocytes and immune cells, but also blood-derived molecules such as immunoglobulins G, complement C3, albumin, thrombin, or fibrin. An activation of astrocytes and microglia has also been reported. Degeneration of endothelial cells and pericytes has been observed, with a decrease of capillary pericytes coverage. The basement membrane was found thickened with observation of collagen IV deposits in humans. Other reported alterations include downregulation of tight junction’s proteins, upregulation of P-glycoprotein, Breast Cancer Resistance Protein, and Aquaporin 4, and formation of extracellular edema. Whether motoneuron degeneration is linked to these alterations still require further investigations as detailed below. AQP4: aquaporin 4; BM: basement membrane ECs: endothelial cells; PCs: pericytes; RBC: red blood cells; TJs: tight junctions. Created with BioRender.com.

**Figure 3 jpm-12-01071-f003:**
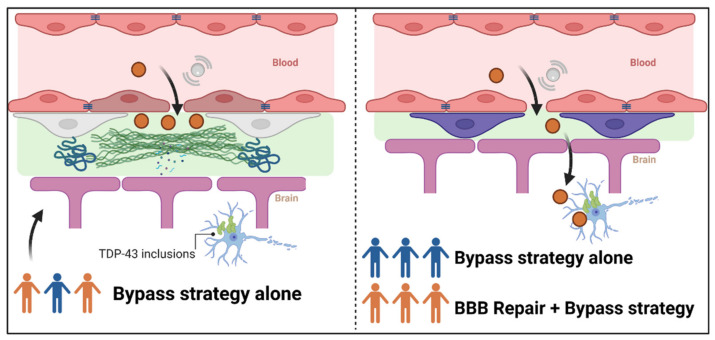
Interest of restoring BBB alterations to optimize the administration of therapeutic molecules. The left panel represents the distribution into the brain of a molecule with transitory opening of the tight junction proteins via microbubble-associated focused ultrasound. In patients who display BBB alterations (orange patients), the deposit of collagen IV and thickening of basement membrane, for example, might prevent the drug from reaching its target in the degenerating motoneurons presenting TDP-43 inclusions. In the right panel, patients have been stratified according to their BBB integrity: intact (blue patients) or disrupted (orange patients). Combination of the BBB repair and a bypass strategy for patients with a disrupted BBB could lead to the disappearance of deposits and membrane thickening and allow the molecule to reach its target, similarly to blue patients with intact BBB.

**Table 1 jpm-12-01071-t001:** Summary of strategies to bypass the BBB/BSCB.

Method	Advantages	Disadvantages
	Mode of administration	
Intrathecal injection	Clinically applicable, various therapeutics	Highly invasive, distribution limited near the injection site
Convection-enhanced delivery	Clinically applicable, various therapeutics, pressure-driven delivery	Highly invasive (surgical procedure), distribution limited
Intranasal administration	Non-invasive	Variability, reduction of efficiency with molecular weight
	Drug modification	
Lipidization	Non-invasive	For water-soluble molecules, rapid elimination
Receptor-mediated transcytosis	Non-invasive, highly specific	Potential toxicity by interference with endogenous ligand
Carrier-Mediated transcytosis	Non-invasive, highly specific	Limited to small molecules
Nanoparticles	Non-invasive, variety of carriers, various therapeutics	Technically challenging, rapid degradation
Neurotropic viruses	Delivery of genes to specific sites in the CNS	Often combined with invasive mode of administration, currently limited to gene therapy, risk of autoimmunity
Neurotropic cells	Delivery of RNA, peptides, proteins or nanoparticles to specific sites in the CNS	Potential toxicity
	BBB/BSCB modifications	
Osmotic disruption	Clinically applicable, various therapeutics	Potential entry of blood neurotoxic compounds
Tight junction downregulation	Various therapeutics	Potential entry of blood neurotoxic compounds, translation to humans limited
Efflux transporter downregulation	Non-invasive	Limited to substrates of efflux transporters, potential toxicity
Focused ultrasounds	Various therapeutics, target of specific sites	Potential entry of blood neurotoxic compounds

**Table 2 jpm-12-01071-t002:** Findings of BBB/BSCB alterations in animal models and humans.

	Animal Findings		Human Findings	
Parameter	Result	References	Result	References
Ultrastructure	Degeneration of ECs, BM thickening, extracellular edema	[1,2,3]	Degeneration of ECs, BM thickening, collagen IV accumulation, extracellular edema	[17]
Cells infiltration	Erythrocytes infiltration	[1]	Erythrocytes infiltration	[18]
	Immune cells infiltration	[1,4,5,6,7]	Immunes cells infiltration	[19,20]
Entry of blood components	IgG deposits	[4,5,8]	IgG deposits	[17,21,22]
	Hemosiderin deposits	[4,5,8]	Hemosiderin deposits	[18]
	Fibrin deposits	[2]	Fibrin deposits	[17,18]
			Hemoglobin deposits	[18,23]
			Thrombin deposits	[18]
Astrocytes	Astrocytosis	[5,7,9]		
	Endfeet degeneration	[3,10,11]	Endfeet degeneration	[11]
Microglia	Microgliosis	[1,2,5,6,9]	Microgliosis	[22,24]
Pericytes	↑ PDGFRβ	[2,5]	Loss of pericytes	[17,18,25]
TJs	↓ mRNA expression	[3]	↓ mRNA expression	[26]
	↓ protein expression	[2,4,12]	↓ protein expression	[17]
			No variation of expression	[23]
	Structurally normal (TEM)	[1]	Structurally normal (TEM)	[17]
	Disruption of TJs (TEM)	[2]		
Efflux transporter	↑ P-gp expression and functionality	[13,14]	↑ P-gp expression	[13,27]
	↑ BCRP expression	[13]	↑ BCRP expression	[13,27]
	No modification BCRP expression	[14]		
Aquaporins	↑AQP4 expression	[6,8,15]	↑ AQP4 expression	[8]
Circulant markers			↑ QAlb, QIgG CSF TP, CSF IgG CSF albumin, CSF hemoglobin in some ALS patients	[23,28,29,30,31,32]
			Association with disease progression	[32]
			No association with disease progression	[28]
Onset of BBB disruption	Presymptomatic stage	[2,11,12]		
	After apparition of symptom	[3,9,16]		
Tracer leakage	Sodium fluorescein	[16]		
	Evans blue	[10,15]		

↑: increased; ↓: decreased; AQP4: aquaporin 4; BCRP: breast cancer resistance protein; BM: basement membrane; ECs: endothelial cells; mRNA: messenger ribonucleic acid; PDGFRβ: platelet-derived growth factor receptor beta; P-gp: P-glycoprotein; QAlb: quotient albumin, QIgG: quotient immunoglobulins G; TEM: transmission electronic microscopy; TJs: tight junctions.

## Data Availability

Not applicable.

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
