# Peer review of "Taking Advantages of Blood–Brain or Spinal Cord Barrier Alterations or Restoring Them to Optimize Therapy in ALS?"

_jpm, 2022, doi:10.3390/jpm12071071_

Round 1

Reviewer 1 Report

In the review article entitled Taking advantages of blood-brain or spinal cord barriers alterations or restoring them to optimize therapy in ALS?", Alarcan and colleagues have collected and complied a large number of updated information on BBB as well as BSCB, providing an important insight into our understanding of ALS pathogenesis and prospective therapeutics. However, there are still a number of concerns that should be addressed.

1) The authors use many “Question Marks: ? ” not only the main title but also the section titles: e.g., sections 3.1, 3.1.1, 3.1.2, 5, and 6. The readers could understand the intentions of the authors, but the reviewer think that too much “?” increases the ambiguity of the sentences. I recommend using more explicit written expressions.

2) Throughout the manuscript, the authors discuss the disease: i.e., ALS; especially the pathobiological aspects obtained from studies on human patients and animal models in a mixed manner. However, it has not still been appreciated that pathobiology of ALS in human and animal models is the same. Thus, the authors must separately explain the aspects and/or findings of human disease and animal models of disease.

3) The sectioning of paragraphs is not clearly categorized. For example, sections 2.1 and 2.2 describe the biological aspects of the BBB, whereas sections from 2.3 (including 2.3.1-2.3.3) are focused on the methodologies and/or tools to analyze the BBB, even though these subsections are included within the same 2nd section. Such sectioning of paragraphs might interrupt the flow of the contexts.

4) Most of the titles of the sections are “nonsentence”, but some are “sentence”: e.g., sections  3.1.3, 3.1.4, 3.2.2, 3.2.4, and 4.1. The reviewer recommend using the consistent style for the section titles.

5) The content in the section 3.2.3 is mostly the perspective views of the authors. However, sections before and after: 3.2.2 and 3.2.4, are described the biological findings. The organization of this section should be rearranged to make the flow of the context consistent.

Author Response

Reviewer 1

In the review article entitled “Taking advantages of blood-brain or spinal cord barriers alterations or restoring them to optimize therapy in ALS?", Alarcan and colleagues have collected and complied a large number of updated information on BBB as well as BSCB, providing an important insight into our understanding of ALS pathogenesis and prospective therapeutics. However, there are still a number of concerns that should be addressed.

Point 1: The authors use many “Question Marks: ? ” not only the main title but also the section titles: e.g., sections 3.1, 3.1.1, 3.1.2, 5, and 6. The readers could understand the intentions of the authors, but the reviewer think that too much “?” increases the ambiguity of the sentences. I recommend using more explicit written expressions.

Response 1: We thank the reviewer for his relevant comment. As recommended, we reduced the use of questions marks and keep it only for the title and the conclusion (see section 4.1, 4.1.1, p7; 4.1.2, p7; section 5.1, p15 ; section 6, p17).

Point 2: Throughout the manuscript, the authors discuss the disease: i.e., ALS; especially the pathobiological aspects obtained from studies on human patients and animal models in a mixed manner. However, it has not still been appreciated that pathobiology of ALS in human and animal models is the same. Thus, the authors must separately explain the aspects and/or findings of human disease and animal models of disease.

Response 2: We thank the reviewer for his remark. Most of the pathophysiological mechanisms observed in animal models are also reported in patients. However, according to the successive failures to find new therapeutics in ALS (with sometimes a benefit effect of some drugs in mice models and no effect in patients), we agree that we need to distinguish the 2 types of findings. Regarding this comment, we reorganized the section about the BBB disruption and the pathogenesis of ALS (section 4.1, p7-8) to separate the findings in humans and in animal models. We also added a table including the comment of the reviewer 2 with a clear separation between humans and animals (page 11-12).

Point 3: The sectioning of paragraphs is not clearly categorized. For example, sections 2.1 and 2.2 describe the biological aspects of the BBB, whereas sections from 2.3 (including 2.3.1-2.3.3) are focused on the methodologies and/or tools to analyze the BBB, even though these subsections are included within the same 2nd section. Such sectioning of paragraphs might interrupt the flow of the contexts.

Response 3: We understand the reviewer remark and we have reorganized some sections to hopefully keep a better flow: we reorganized the sections about the pathophysiological aspects of the BBB in ALS (section 4.1, p7-8) and about the impact on drug pharmacokinetics (section 4.2, p9-14) in accordance with the reviewer points 2 and 5, respectively. We have separated the tools to analyze the BBB (section 3, p5-6) from the rest of the section 2.

Point 4: Most of the titles of the sections are “nonsentence”, but some are “sentence”: e.g., sections  3.1.3, 3.1.4, 3.2.2, 3.2.4, and 4.1. The reviewer recommend using the consistent style for the section titles.

Response 4: We replace “sentence” titles to “nonsentence” titles (section 4.1.3, p8; section 4.2.2, p9; section 4.2.3, p10; section 5.1, p14) according to the reviewer recommendation.

Point 5: The content in the section 3.2.3 is mostly the perspective views of the authors. However, sections before and after: 3.2.2 and 3.2.4, are described the biological findings. The organization of this section should be rearranged to make the flow of the context consistent.

Response 5: We thank the reviewer for his relevant remark and we have rearranged this section to describe the biological findings before our perceptive views (section 4.2, p9-14).

Reviewer 2 Report

The authors have done excellent work in this review.  I suggest adding a Table to summarize the main points with relative references.

Author Response

Reviewer 2

Comments and Suggestions for Authors

The authors have done excellent work in this review. 

Point 1: I suggest adding a Table to summarize the main points with relative references.

Response 1: We are pleased to see that the reviewer appreciates our review. We added the following table (Please see the attachment) containing the main alterations of the BBB/BSCB found in animal or human studies (table 2, page 11-12). We agree that this type of table would be directly usable by the readers.

Reviewer 3 Report

In their review article “Taking advantages of blood-brain or spinal cord barriers alterations or restoring them to optimize therapy in ALS?”, Alarcan and colleagues provide a comprehensive and well-written state-of-the-art overview on the (yet limited) data of BBB/BSCB alterations in motor neurone disease. In my opinion, the topic is of high relevance and the review falls within the scope of J Pers Med.

However, some major revisions need to be considered before publication:

Introduction of a table comprising the different modes of BBB / BSCB penetration for the different pharmaceutical approaches is recommended to further clarify the topic.

ALS is a heterogeneous disorder, not only regarding genetics, but also localization and progression of symptoms  – the authors should include thoughts on specific BBB/BSCB sites to be targeted in bulbar vs. spinal onset ALS.

Small molecules and nanoparticles (e.g., PMID  31926785) might be the most promising mechanism to transcend the BBB / BSCB in general, but their discussion seems underrepresented in the manuscript.

When discussing focused ultrasound, the ALS study in Nat Commun should be critically included (PMID  31558719).

In the review on the limited action of ASOs (l. 540 ff), the authors should include the findings of the phase 2 study of tofersen which indeed showed target engagement (SOD1 reduction) upon treatment, also in the cited phase 3 study.

Minor comments:

Fig. 2: “BM” should be explained in the legend.

l. 222: “than”

l. 369/370: “increased” (two times)

l. 372: verb is missing

Author Response

Reviewer 3

In their review article “Taking advantages of blood-brain or spinal cord barriers alterations or restoring them to optimize therapy in ALS?”, Alarcan and colleagues provide a comprehensive and well-written state-of-the-art overview on the (yet limited) data of BBB/BSCB alterations in motor neuron disease. In my opinion, the topic is of high relevance and the review falls within the scope of J Pers Med.

 However, some major revisions need to be considered before publication:

 Point 1: Introduction of a table comprising the different modes of BBB / BSCB penetration for the different pharmaceutical approaches is recommended to further clarify the topic.

Response 1: We thank the reviewer for his remark and added the following table (Please see the attachment) with the different modes of BBB/BSCB penetration in section 2.2 (table 1, page 4-5). We agree that this table would be useful for the readers.

Point 2: ALS is a heterogeneous disorder, not only regarding genetics, but also localization and progression of symptoms  – the authors should include thoughts on specific BBB/BSCB sites to be targeted in bulbar vs. spinal onset ALS.

Response 2: We are grateful to the reviewer for his relevant comment. We included information to comment that the localization of BBB alterations might not coincide with that of motoneuron loss which is different between spinal and bulbar onset (section 4.2.4, p13, l450-454): “The localization of primally affected motoneurons is different between patients with spinal onset (muscle weakness of the limbs) and patients with bulbar onset (dysarthria, dysphagia, speech difficulty) [2]. The localization of the increased BBB permeability reported in ALS might not be associated with that of affected motoneurons in spinal or bulbar forms.”

 We have also commented the fact that the magnetic resonance guided ultrasounds might give the opportunity to specifically target the area of interest for bulbar or spinal forms (section 6, p16, l676-678): “ Moreover, targeting specific regions of the CNS with MR-guided focused ultrasounds could allow specific delivery of the therapeutic agents to the affected motoneurons ac-cording to the onset of the disease (spinal or bulbar).”.

Point 3: Small molecules and nanoparticles (e.g., PMID  31926785) might be the most promising mechanism to transcend the BBB / BSCB in general, but their discussion seems underrepresented in the manuscript.

Response 3: We thank the reviewer for his remark. Indeed, some small lipid-soluble molecules are able to cross the BBB/BSCB through passive diffusion which is valuable for the treatment of brain disease. However, as clinical trials on ALS patients continue to fail, it could suggest that the amount of drug reaching the target is still insufficient and that a strategy to enhance the crossing of the BBB/BSCB is needed. Nanoparticles is indeed a promising mechanism to overcome these barriers, but also to monitor the localization of the therapeutic and release it close to its target. We have briefly added a list of different types of nanoparticles in the section 2.2 (p4, l164-166) and added an example of the use of inorganic nanoparticle, cerium oxide, evaluated in ALS rodent (section 6, p17, l641-647). We have also included the article proposed by the reviewer (section 6, p17, l41).

Point 4: When discussing focused ultrasound, the ALS study in Nat Commun should be critically included (PMID  31558719).

 Response 4:  The study of Abrahao et al. was included in the section 4.1.2 ( p8, l285-289) As suggested wisely by the reviewer, we have also mentioned it in the section about the strategies to overcome the BBB attempted in ALS (section 6, p17, l672-679) to support the fact that this study could be a turning point in ALS :

“safety and feasibility of transitory opening of the BBB using MR-guided focused ultra-sounds in ALS has been demonstrated by Abraho et al [33]. After optimization in future trials, this method could represent a non-invasive opportunity to deliver various therapeutic agents into the CNS, including small molecules and nanoparticles. Moreover, targeting specific regions of the CNS with MR-guided focused ultrasounds could allow specific delivery of the therapeutic agents to the affected motoneurons according to the onset of the disease (spinal or bulbar). To note, application of this method to the spinal cord will be challenging in primates vertebrae [107]”.  

Point 5: In the review on the limited action of ASOs (l. 540 ff), the authors should include the findings of the phase 2 study of tofersen which indeed showed target engagement (SOD1 reduction) upon treatment, also in the cited phase 3 study.

Response 5: We included these findings as suggested by the reviewer (section 6, p17, l654-656):

 “intrathecal administration of an antisense oligonucleotide (ASO) targeting SOD1 (Tofersen) recently failed a first phase III clinical trial [100], despite promising results in rodents [101] and signs of target engagement with a 30% reduction level of CSF SOD1 for the maximum tested dose (100 mg) in phase II trial [102]”

Point 6: Minor comments:

Fig. 2: “BM” should be explained in the legend.

  1. 222: “than”
  2. 369/370: “increased” (two times)
  3. 372: verb is missing

Response 6:  We thank the reviewer and we have made the corrections in consequence

Round 2

Reviewer 1 Report

In the revised manuscript entitled Taking advantages of blood-brain or spinal cord barriers alterations or restoring them to optimize therapy in ALS?", the authors appropriately improved the context of the manuscript by responding to my original concerns. There are no further comments to this manuscript.

Reviewer 3 Report

The authors have addressed all comments by the reviewers sufficiently.

Language style of the newly added sections and tables needs to be double checked because there is bad wording in many sections.